# The Impact of the SARS-CoV-2 Outbreak on the Psychological Flexibility and Behaviour of Cancelling Medical Appointments of Italian Patients with Pre-Existing Medical Condition: The “ImpACT-COVID-19 for Patients” Multi-Centre Observational Study

**DOI:** 10.3390/ijerph18010340

**Published:** 2021-01-05

**Authors:** Giuseppe Deledda, Niccolò Riccardi, Stefania Gori, Sara Poli, Matteo Giansante, Eleonora Geccherle, Cristina Mazzi, Ronaldo Silva, Nicoletta Desantis, Ambra Mara Giovannetti, Alessandra Solari, Paolo Confalonieri, Licia Grazzi, Elena Sarcletti, Gabriella Biffa, Antonio Di Biagio, Carlo Sestito, Roland Keim, Alida M. R. Di Gangi Hermis, Mariantonietta Mazzoldi, Alessandro Failo, Anna Scaglione, Naida Faldetta, Patrizia Dorangricchia, Maria Moschetto, Hector Josè Soto Parra, Jennifer Faietti, Anna Di Profio, Stefano Rusconi, Andrea Giacomelli, Fabio Marchioretto, Filippo Alongi, Antonio Marchetta, Giulio Molon, Zeno Bisoffi, Andrea Angheben

**Affiliations:** 1Clinical Psychology Unit, IRCCS Sacro Cuore Don Calabria Hospital, Negrar di Valpolicella, 37024 Verona, Italy; sara.poli@sacrocuore.it (S.P.); matteo.giansante@sacrocuore.it (M.G.); eleonora.geccherle@sacrocuore.it (E.G.); 2Department of Infectious—Tropical Diseases and Microbiology, IRCCS Sacro Cuore Don Calabria Hospital, Negrar di Valpolicella, 37024 Verona, Italy; niccolo.riccardi@sacrocuore.it (N.R.); zeno.bisoffi@sacrocuore.it (Z.B.); andrea.angheben@sacrocuore.it (A.A.); 3Oncology Department, IRCCS Sacro Cuore Don Calabria Hospital, Negrar, 37024 Verona, Italy; stefania.gori@sacrocuore.it; 4Clinical Research Unit, IRCCS Sacro Cuore Don Calabria Hospital, Negrar di Valpolicella, 37024 Verona, Italy; cristina.mazzi@sacrocuore.it (C.M.); rolando.silva@sacrocuore.it (R.S.); nicoletta.desantis@sacrocuore.it (N.D.); 5Unit of Neuroepidemiology, Fondazione IRCCS Istituto Neurologico Carlo Besta, 20145 Milan, Italy; Ambra.Giovannetti@istituto-besta.it (A.M.G.); Alessandra.Solari@istituto-besta.it (A.S.); 6Multiple Sclerosis Centre, Neuroimmunology Unit, Clinical Neurology Department, Fondazione IRCCS Istituto Neurologico Carlo Besta, 20145 Milan, Italy; paolo.confalonieri@istituto-besta.it; 7School of Psychology, Faculty of Health and Behavioural Sciences, University of Queensland, Brisbane, 4072 QLD, Australia; 8Headache Center, Neurology Department, Fondazione IRCCS Istituto Neurologico Carlo Besta, 20145 Milan, Italy; Licia.Grazzi@istituto-besta.it; 9Clinical Psychology and Psychotherapy Unit, IRCCS Ospedale Policlinico San Martino, 16132 Genoa, Italy; elena.sarcletti@hsanmartino.it (E.S.); gabriella.biffa@hsanmartino.it (G.B.); 10Infectious Diseases Clinic, Policlinico San Martino Hospital-IRCCS, 16132 Genoa, Italy; antonio.dibiagio@hsanmartino.it; 11Infectious Diseases Clinic, Department of Health Sciences, University of Genoa, 16132 Genoa, Italy; 12Service of Psycho Oncology, San Giovanni di Dio Hospital, Azienda Sanitaria Provinciale di Crotone, 88900 Crotone, Italy; pnlcalabria@tiscali.it; 13General Hospital Bressanone, Azienda Sanitaria dell’Alto Adige, Bressanone, 39042 Bolzano, Italy; roland.keim@sabes.it (R.K.); alida.digangi@sabes.it (A.M.R.D.G.H.); 14General Hospital Bolzano Azienda Sanitaria dell’Alto Adige, 39100 Bolzano, Italy; mariantonietta.mazzoldi@sabes.it (M.M.); alessandro.failo@outlook.it (A.F.); 15CFU-Italia ODV (Fybromyalgia Association), Castenaso, 40085 Bologna, Italy; anna.scaglione1@virgilio.it; 16Department of Oncoplastic Surgery of Breast Unit. V. Cervello Hospital, 90100 Palermo, Italy; naida.faldetta@gmail.com (N.F.); patriziadorangricchia@gmail.com (P.D.); 17Oncology Unit, Azienda Ospedaliero Universitaria (A.O.U.) Policlinico Vittorio Emanuele, 95123 Catania, Italy; mariamoschetto23@yahoo.it (M.M.); hsotoparra@yahoo.it (H.J.S.P.); 18Cardiac Surgery Unit, Salus Hospital GVM Care & Research, 42123 Regio Emilia, Italy; psicologia-sal@gvmnet.it; 19Clinical Oncology Unit, S.S. Annunziata Hospital, 66100 Chieti, Italy; annadiprofio@gmail.com; 20Infectious Diseases Unit, Department of Biomedical and Clinical Sciences (DIBIC) Luigi Sacco, University of Milan, 20157 Milan, Italy; stefano.rusconi@unimi.it (S.R.); andrea.giacomelli@unimi.it (A.G.); 21Neurological Unit, IRCCS Sacro Cuore Don Calabria Hospital, Negrar di Valpolicella, 37024 Verona, Italy; fabio.marchioretto@sacrocuore.it; 22Advanced Radiation Oncology Department, Sacro Cuore Don Calabria Hospital, Negrar di Valpolicella, 37024 Verona, Italy; filippo.alongi@sacrocuore.it; 23University of Brescia, 25121 Brescia, Italy; 24Rheumatology Unit, IRCCS Sacro Cuore Don Calabria Hospital, Negrar di Valpolicella, 37024 Verona, Italy; antonio.marchetta@sacrocuore.it; 25Cardiology Deparment, IRCCS Sacro Cuore Don Calabria Hospital, Negrar di Valpolicella, 37024 Verona, Italy; giulio.molon@sacrocuore.it

**Keywords:** COVID-19, virus, pandemic, lockdown, psychological flexibility, depression, anxiety, stress, psychological impact, cancelling medical appointments

## Abstract

Psychological distress imposed by the SARS-CoV-2 outbreak particularly affects patients with pre-existing medical conditions, and the progression of their diseases. Patients who fail to keep scheduled medical appointments experience a negative impact on care. The aim of this study is to investigate the psychosocial factors contributing to the cancellation of medical appointments during the pandemic by patients with pre-existing health conditions. Data were collected in eleven Italian hospitals during the last week of lockdown, and one month later. In order to assess the emotional impact of the SARS-CoV-2 outbreak and the subject’s degree of psychological flexibility, we developed an ad hoc questionnaire (ImpACT), referring to the Acceptance and Commitment Therapy (ACT) model. The Impact of Event Scale-Revised (IES-R), the Depression, Anxiety and Stress Scale (DASS) and the Cognitive Fusion Questionnaire (CFQ) were also used. Pervasive dysfunctional use of experiential avoidance behaviours (used with the function to avoid thought, emotions, sensations), feelings of loneliness and high post-traumatic stress scores were found to correlate with the fear of COVID-19, increasing the likelihood of cancelling medical appointments. Responding promptly to the information and psychological needs of patients who cancel medical appointments can have positive effects in terms of psychological and physical health.

## 1. Introduction

On 21 February 2020, the first case of COVID-19, as caused by the SARS-CoV-2 virus was confirmed in Italy [1]. As an epidemic mounted, it became crucial to identify strategies, and source facilities, capable of providing health care without compromising the safety of medical professionals, administrative staff and patients [2].

Hospitals in northern Italy were quickly swamped with high numbers of COVID-19 patients. Facilities and resources had to be hurriedly reorganised, as a matter of urgency, so that dedicated COVID-19 units could be opened. At the end of April 2020, 17.4% of all identified individuals with SARS-CoV-2 were experiencing severe symptoms of COVID-19 that required hospitalisation. Meanwhile, 13.6% of individuals with SARS-CoV-2 were asymptomatic, 17.2% pauci-symptomatic, and 35.7% displayed only mild symptoms. The remaining 16.3% had symptoms for which the severity level was not specified. In addition, of all subjects with COVID-19, 1.9% were hospitalised in critical condition requiring intensive care [3]. Between February and April 2020, 47.3% of cases were males with a median age of 62 years (range 0–100). As of 3 June 2020, 230,811 cases of SARS-CoV-2 had been diagnosed and 32,354 deaths had arisen from COVID-19 in Italy alone [3].

Comorbidity appears to be a determining factor in COVID-19 severity and prognosis. In 34.7% of cases, at least one co-morbidity has been reported, including cardiovascular diseases, respiratory diseases, hypertension, diabetes, immunodeficiencies, metabolic diseases, cancers, obesity, kidney diseases, and other chronic diseases [4]. Hospitals and care centres have adopted various measures in efforts to contain the spread of SARS-CoV-2, especially amongst patients with pre-existing pathologies. Such strategies include the isolation of patients and the postponement of medical appointments, therapy and follow-ups, which are deemed non-essential.

The psychological distress imposed by the SARS-CoV-2 outbreak has been widely reported. This particularly affects patients with pre-existing medical conditions, for example, a large proportion of young cancer patients have expressed their deep concern at being more susceptible to severe complications and feeling the burden of their parents’ worry [5]. Rheumatology patients with systemic autoimmune disease are also highly vulnerable and have reported several concerns including whether or not to continue with immunosuppressive medication [6]. Further to this, pre-existing medical conditions, chronic illness and/or self-evaluated poor health, are considerable risk factors for loneliness, anxiety, depression and psychological distress [7,8,9].

In many cases, patients have been contacted by medical centres to cancel or postpone scheduled consultations, treatments and/or surgery to reduce the risk of contracting SARS-CoV-2. In turn, patients have also contacted medical centres to cancel appointments.

It should be noted that many patients have not cancelled their scheduled medical appointments, despite the impact of the external context, including the many social media information (sometimes very conflicting or confusing) and institutions (ministry of health, medical centers).

During the SARS-CoV-2 outbreak, patients’ reasons for cancelled appointments included being unable to: get time off work, secure childcare, and/or find a safe mode of transport. In addition, patients reported cancelling appointments because their health concern had resolved itself, or because they were too ill to attend [10].

Research conducted prior to the SARS-CoV-2 outbreak indicates that patients who schedule medical appointments and fail to keep them have a negative impact on patient care [11,12,13]. Patients who cancel appointments tend to be younger [12,14,15] and of lower socioeconomic status. They often have a history of psychosocial problems too [16,17].

Thus far, there have been no published studies on the behaviour of cancelling medical appointments of patients with pre-existing medical condition during the SARS-CoV-2 outbreak, and especially none that consider the psychosocial factors influencing the decision to cancel appointments regularly scheduled by medical centres.

The behaviour of cancellation medical appointments can be linked to different psychological functions (e.g., avoiding or controlling an adverse internal context, or, of moving in a more adaptive way based on external contingencies), that occur in relation to the external context (decisions of the ministry, medical centres, social and health contexts) and in relation to the internal context (thoughts, emotions, physical sensations). Therefore, in some cases this behaviour it can be functionally adaptive to the context or in other cases it can be highly dysfunctional to the context and have negative consequence on one’s own and others’ health. The ability to adapt is closely linked to the degree of psychological flexibility, defined as acting in accordance with personal goals and values, in the presence of potentially interfering thoughts and feelings, and with a greater appreciation of what their current situation or context allows [18].

Moreover, recent studies have shown that psychological flexibility mitigated the detrimental impacts of the pandemic on mental health, peritraumatic distress, anxiety, depression, insomnia and facets of psychological inflexibility exacerbated the impact of these risks [19,20,21].

As such, the aim of this study is to investigate the psychosocial factors contributing to the cancellation of medical appointments, by patients with pre-existing health conditions, in Italy, during the SARS-CoV-2 outbreak.

## 2. Materials and Methods

### 2.1. Procedure

We have developed a cross-sectional study in which anonymous questionnaires were implemented to assess the impact of the SARS-CoV-2 outbreak on compliance with medical treatment and on adherence to a healthy lifestyle, amongst patients with pre-existing medical conditions. The questionnaires could be completed as a hard copy or as an electronic version, online. Hospital inpatients and outpatients were the first potential participants to be made aware of the study. A “snowball sampling strategy”, focusing on recruiting other participants with pre-existing medical conditions of mainland Italy, then followed. The study involved eleven Italian hospitals located throughout Italy, including northern, central and southern regions.

Data were collected during the last week of lockdown (20–27 April 2020) (T1 phase), and again one month later (22–30 May 2020) (T2 phase), from individuals aged eighteen years or over with pre-existing medical conditions, and who were not afflicted with COVID-19 at the time of data collection. Potential respondents were made aware of the survey by other study participants, or by reading information about the study that was made available to them during a hospital visit. Potential respondents were invited to complete questionnaires online or as hard copies for patients visiting the hospital. All participants provided informed consent to participate in the anonymous survey. The procedures were clearly explained in writing, and participants could withdraw from the study at any time without explaining their reasons for doing so. Expedited ethics approval was obtained from the Institutional Review Board of participants, the Ethical Committee for Clinical Trials of the Provinces of Verona and Rovigo in Northern Italy (Prog. 2642CESC), and the Local Ethics Committees of Centres that collaborated in this study, which conformed to the principles embodied in the Declaration of Helsinki. Information about this study was posted on a dedicated website of the Clinical Psychological Service of the Scientific Institute for Research Hospitalisation and Health Care (IRCSS) Sacro Cuore Don Calabria Hospital in Negrar.

### 2.2. Sociodemographic Data and Contextual Information

The following sociodemographic and contextual data were collected: gender, age, level of education, residential location, marital status, employment status, economic losses relating to the SARS-CoV-2 outbreak, parental status, family composition, cohabitation status before and during lockdown, compliance with physical isolation regulations, and the type of medical pathology suffered. Respondents were also asked about their sources of information regarding SARS-CoV-2 infection rates as well as COVID-19 symptom management. They were also asked to rate their trust in these sources, and about their main sources of psychological support. Information regarding lifestyle, interpersonal relationships and psychological trauma relating to the SARS-CoV-2 outbreak, (for example, changes in relationships, and the occurrence of any bereavements). Data were also collected regarding physical health status and treatment phase (diagnostics, active therapies, follow-up, palliative care). Data about access to health services during lockdown specifically regarding the location of medical centres, clinic-based consultations, admission to hospital, and distress relating to isolation during lockdown were also collected. Data were also collected regarding mental health status psychiatric history, the emotional impact of the SARS-CoV-2 outbreak, any behaviours stemming from a fear of contracting SARS-CoV-2, and any other lifestyle choices that may impact one’s general health, for example, alcohol consumption or smoking status.

### 2.3. The “ImpACT” Questionnaire

In order to evaluate the behaviour of patients in a stressful situation, in this instance, the SARS-CoV-2 outbreak, we developed the “ImpACT” 31-item self-report questionnaire. The theoretical model of reference used was the functional contextualism and the Relational Frame Theory (RFT) [22] that encompasses the concept of psychological flexibility that underpins the Acceptance and Commitment Therapy model (ACT) [23]. Within ACT, psychological flexibility is conceptualized as a product of six distinct but interrelated sub-processes: acceptance; defusion; self as context; present moment awareness; values; and committed action [24]. ACT is a trans-diagnostic therapeutic approach that conceptualizes psychological suffering as primarily a function of attempts to avoid unwanted private experiences (experiential avoidance) and a resultant or contingent reduction in personally meaningful pursuits (values-inconsistent behaviour) [23]. ACT aims to reduce experiential avoidance (in the service of increasing values-consistent behaviour) by fostering psychological flexibility—“the ability to contact the present moment more fully as a conscious human being, and to change or persist in behavior when doing so serves valued ends” [24].

In general terms cognitive fusion refers to excessive or improper regulation of behaviour by verbal processes, such as rules and derived relational networks. When cognitive fusion increases, human behaviour is less sensitive to environmental contingencies. As a result, people may act in a way that is inconsistent with what the environment affords relevant to chosen values and goals [23].

Experiential avoidance is the attempt to alter the form, frequency, or situational sensitivity of private events even when doing so causes behavioural harm [23]. Due to the temporal and comparative relations present in human language, so-called “negative” emotions are verbally predicted, evaluated, and avoided. Experiential avoidance is based on this natural language process, a pattern that is then amplified by the culture into a general focus on “feeling good” and avoiding pain. Unfortunately, attempts to avoid uncomfortable private events tend to increase their functional importance, both because they become more salient and because these control efforts are themselves verbal linked to conceptualized negative outcomes, and thus tend to narrow the range of behaviours that are possible since many behaviours might evoke these feared private events [24]. Furthermore, the experiential avoidance is the phenomenon that occurs when a person is unwilling to remain in contact with particular private experiences (e.g., bodily sensations, emotions, thoughts, memories, behavioural predispositions) and takes steps to alter the form or frequency of these events and the contexts that occasion them. Occasionally are use terms such as emotional avoidance or cognitive avoidance rather than the more generic experiential avoidance when it is clear that these are the relevant aspects of experience that the person seeks to escape, avoid, or modify [23].

Hayes et al. [25] propose that psychological flexibility can be pragmatically defined in terms of three “dyadic” processes: (1) “openness to experience and detachment from literality” (acceptance; defusion); (2) “self-awareness and perspective taking” (present moment awareness; self as context); and (3) “motivation and activation” (values; committed action).

In the ImpACT questionnaire, the word “COVID-19” was used instead of “stressful situation”, in order to identify specific contextual behavioural responses (Appendix A).

### 2.4. Measurements Made Using the Impact of Event Scale-Revised (IES-R), the Depression, Anxiety and Stress Scale (DASS-21) and the Cognitive Fusion Questionnaire (CFQ)

The traumatic and stressful psychological impact of COVID-19 was measured using the Impact of Event Scale-Revised (IES-R) [26]. The IES-R is a 22-item self-report questionnaire that evaluates the degree of emotional impact of a traumatic event and the presence of probable post-traumatic stress disorder (PTSD). The total IES-R score was divided into: 0–23 (normal), 24–32 (mild psychological impact), 33–36 (moderate psychological impact), and >37 (severe psychological impact) [27]. A score above 50 indicates a probable case of PTSD [28]. The Italian version of the IES-R [29] has a clear factor structure with three independent and robust dimensions: intrusion, avoidance, and hyperarousal. The IES-R has previously been used in SARS-CoV-2 related research [7,30]. In our study, IES-R showed good internal consistency in both rounds of questionnaire (0.94 in T1 and 0.95 in T2).

Depression, anxiety and stress, defined as irritability, nervous tension, difficulty relaxing, and agitation, were measured using the Depression, Anxiety and Stress Scale (DASS-21) self-report questionnaire [31]. The total depression subscale score was divided into: normal (0–9), mild depression (10–12), moderate depression (13–20), severe depression (21–27), and extremely severe depression (28–42). Questions 2, 4, 7, 9, 15, 19, and 20 formed the anxiety subscale. The total anxiety subscale score was divided into normal (0–6), mild anxiety (7–9), moderate anxiety (10–14), severe anxiety (15–19), and extremely severe anxiety (20–42). Questions 1, 6, 8, 11, 12, 14, and 18 formed the stress subscale. The total stress subscale score was divided into normal (0–10), mild stress (11–18), moderate stress (19–26), severe stress (27–34), and extremely severe stress (35–42). Cronbach’s alpha for the total scales was 0.96 in both T1 and T2.

Cognitive fusion, described as the degree to which an individual becomes caught up in their thoughts, was measured using the Cognitive Fusion Questionnaire (CFQ) [32]. Learning based on verbal processes is argued to play a role in the development of mental health difficulties [33]. The CFQ is a 7-item self-administered questionnaire that has shown good internal consistencies in its original and Italian validations. Higher scores reflect a higher degree of cognitive fusion. The test showed good internal consistency, with Chronbach’s alpha of 0.91 in T1 and T2.

### 2.5. Statistical Analysis

All analyses were performed on patients with pre-existing medical conditions using R software, version 3.6.1 [34]. All 31 items of the ImpACT questionnaire were dichotomized merging the Likert scale items from 1 “rarely true” to 5 “always true” in one item: “true”. Demographic characteristics were summarised by means of descriptive statistics and frequency distributions. Patient related factors (e.g., demographic, knowledge and concern-related, and health related factors), scores of the IES-R, DASS-21 and CFQ questionnaire and the 31 items of the ImpACT questionnaire, were analysed by univariable logistic regression models to explore their association with the likelihood of cancelling a medical appointment due to fear of COVID-19 (ImpACT question eleven: “I cancelled an appointment with my specialist doctor due to fear of COVID-19.”). Only variables significantly associated (*p*-value < 0.2) to cancelling an appointment with a specialist were included in the full logistic regression model. Model-building strategies included checking for convergence, correlation and goodness-of-fit test. The likelihood-ratio test was used to compare candidate models. A statistical significance level of 0.05 was adopted for all tests.

## 3. Results

### 3.1. Participants

Among 884 respondents in the first round of questionnaires, 126 were excluded because did not fulfil the inclusion criteria (34 have had psychological disorders, 73 were defined themselves as healthy and 19 did not declare any health condition). Out of 817 respondents during the second round of questionnaires, 119 were excluded because of presence of exclusion criteria (26 have had psychological disorders, 74 defined themselves as healthy and 19 did not declare any health condition). Therefore, the final sample consisted of 758 patients with pre-existing medical conditions in T1 and 698 in T2.

The sociodemographic characteristics of these sample from the lockdown phase (T1) and the post lockdown phase (T2) are shown in Table 1. There were 25.2% male in T1 and 37.0% in T2. The mean age of participants was 50 years in T1 and 52 years in T2. In the first phase, 65.2% of patients declared to be married or in a relationship and 82.1% held a high-school degree or higher degree of education. In the second phase, the individuals married or in a relationship were 63.2% and the ones who held a high-school degree or higher degree of education were 80.7%. There were 60.6% parents in T1 and 60.7% in T2. A total of 29.8% subjects were employed but working remotely at the time of the first survey and 28.3% at the time of the second survey. Respondents form Northern Italy were 74.8% in the lockdown phase and 74.1% in the post lockdown phase.

Some of the reported pre-existing medical conditions were: cancer (27.0% in T1 and 33.2% in T2), rheumatic disease (15.8% in T1 and 8.2% in T2), multiple sclerosis (13.7% in T1 and 10.4% in T2), cardiovascular disease (9.9% in T1 and 6.3% in T2) and HIV (3.8% in T1 and 11.5% in T2). Finally, 72.0% subjects reported the official websites of national institutions as being the most reliable source of information regarding SARS-CoV-2 and COVID-19 during the first round of questionnaire and 63.8% in the second. This was followed by doctors, nurses or psychologists (44.3% in T1 and 46.4% in T2), and online newspapers (25.9% in T1 and 24.6% in T2). Respondents considered social media (19.0% in T1 and 13.3% in T2) and instant messaging apps (7.5% in T1 and 5.9% in T2) to be trusted sources of information as well.

Data regarding the psychological impact of the SARS-CoV-2 outbreak on individuals with pre-existing medical conditions are shown in Table 2. In the T1 lockdown phase of the study, 19.4% of subjects reported severe to extremely severe stress symptoms, as determined using the DASS-21 Stress subscale, and 28.9% reported moderate symptoms. This corresponded to 21.6% and 22.8% in the post lockdown T2 phase of the study. Concerning anxiety, 15.6% of participants in the T1 phase reported severe to extremely severe anxiety and 17.8% reported moderate symptoms, as determined using the DASS-21Anxiety subscale. This corresponded to 13.5% and 13.6% in the T2 phase. As for the depression subscale, 13.0% participants reported severe to extremely severe depression in the T1 phase and 19.6% reported moderate depression. This corresponded to 13.1% and 16.8% in the T2 phase. In addition, we found that in T1 and T2 phases, subjects who reported severe post-traumatic symptoms were respectively 33.4% and 33.0%. The median total score collected from CFQ data, was 20 in T1 and 21 T2.

Further to this, 35.2% individuals stated to have cancelled an appointment with their specialist doctor due to fear of COVID-19 in the lockdown phase and 29.4% in the post lockdown phase.

### 3.2. Sociodemographic Variables and Their Effect on Cancelling Medical Appointments

Results from the multivariate analysis show that during the lockdown (T1), married (OR = 2.63; 95% CI = 1.24–5.70) and widowed subjects (OR = 5.19; 95% CI = 0.99–25.90) were more likely to cancel a medical appointment than single subjects. However, age, gender, level of education and employment status were not associated with compliance to medical appointments. Further to this, after lockdown (T2), no differences were found between the various sociodemographic subgroups regarding compliance with medical appointments.

### 3.3. Psychological Health Status and Its Effect on Cancelling Medical Appointments

Analysis of the lockdown (T1) phase revealed a significant association between post-traumatic stress symptoms (IES-R scores) and the likelihood of cancelling a medical appointment (OR = 1.02; 95% CI = 1.00–1.05). This association was not apparent in the second, post-lockdown (T2) phase.

Other psychological variables, including depression, anxiety, and stress (DASS-21), were not associated cancelling medical appointments (Table 3). 

### 3.4. The Relationship with Health Care Professionals and Its Effect on Cancelling a Medical Appointment

The likelihood of a patient cancelling a medical appointment if they had spoken with a specialist over the phone during the lockdown (T1) was 2.24 times higher than if they had not spoken to a specialist (95% CI = 1.28–3.95). Similar results were found for subjects interviewed after the lockdown (T2) whereby patients that had spoken to a specialist were 2.13 times more likely to cancel a medical appointment (95% CI = 1.16–3.97) (Table 3).

In addition, the concern that medical staff would develop COVID-19 and be too unwell to be able to treat subjects effectively, was significantly associated with cancelling appointments in both phases. Patients who were worried about encountering contagious medical staff, were 2.17 times more likely to cancel a medical appointment during T1 phase (95% CI = 1.25–3.82) and 3.41 times more likely to do so during T2 phase (95% CI = 1.83–6.51) than patients that did not have this concern.

### 3.5. Isolation and Loneliness and Their Effect on Cancelling a Medical Appointment

During the lockdown phase (T1), patients who felt more alone than usual showed a significantly higher likelihood of cancelling a medical appointment as opposed to patients who did not feel alone (OR = 1.96; 95% CI = 1.06–3.68). Further to this, subjects who kept in contact with fellow patients were 2.13 times more likely to cancel their medical appointment than those who were not in contact with other patients (95% CI = 1.25–3.64) (Table 3).

### 3.6. Dysfunctional Use of Experiential Avoidance during the SARS-CoV-2 Outbreak and Their Effect on Cancelling a Medical Appointment

A significant association was observed amongst patients regarding increased food intake, increased alcohol consumption and/or increased numbers of cigarettes smoked, as coping strategies to manage emotions such as, boredom, apathy, and/or sadness. The likelihood of a subject that uses such strategies would cancel a medical appointment was 1.73 times higher than for subjects that did not adopt such strategies during lockdown (T1) (95% CI = 1.03–2.90) and 2.57 times higher post lockdown (T2) (95% CI = 1.43–4.68).

## 4. Discussion

Our results indicate that the SARS-CoV-2 outbreak, and resultant lockdown, have had a significant psychological impact on individuals with pre-existing medical conditions. About a third of the sample reported severe post-traumatic symptoms, 19.4% severe to extremely severe stress symptoms, about 15% reported severe to extremely severe anxiety, and 13% severe to extremely severe depression (during lockdown (T1) and similar data after (T2)).

Psychological distress appears to be linked to a fear of contracting SARS-CoV-2 and developing COVID-19, which has affected medical appointment attendance and disrupted disease management approaches. Fear of COVID-19 was apparent in many subjects in both phases of the study (during lockdown and after) and was found to correlate with high post-traumatic stress scores. Of note, subjects with HIV did not show any greater propensity to cancel medical appointments due to concerns about COVID-19 as compared to subjects with cancer, rheumatic disease, multiple sclerosis, chronic migraine, cardiovascular disease, endocrine diseases, gynaecological disorders, or others (such as gastrointestinal diseases, immune disorders and infectious diseases).

Failure to attend medical appointments is a problem commonly encountered by clinicians in an ambulatory setting. Other than the obvious disruption to working practices, and waste of resources, a patient’s disease management can also be adversely affected by such behaviour [35]. From our results, psychological variables including, depression, anxiety and stress, were not associated with a failure to attend medical appointments. Conversely, as we expected, we found a significant association between post-traumatic stress symptoms (evaluated by IES-R questionnaire) and the tendency of cancelling a medical appointment during lockdown (T1); probably this data reflect the stress of the sudden change to everyday life that was induced by the lockdown. In addition, while DASS-21 questionnaire is a more general test of anxiety, stress and depression, IES-R questionnaire is more sensible to this context. The latter can therefore better evaluate people’s reaction to this potentially traumatic, stressful event which in turn can influence people behaviour. Finally, for the other psychological variables (depression, anxiety and stress) there were no apparent differences between the two time periods studied (during lockdown, and after). Moreover, the type of pathology or sociodemographic variable (including age, gender, level of education, and employment status) were not associated with the cancellation of medical appointments.

In summary, during the lockdown, some factors accounted for an increased likelihood in cancelling a medical appointment: the psychological impact of the SARS-CoV-2 outbreak as measured by IES-R, marital status (married, in a relationship or widowed), feelings of loneliness, and a set of different behavioural patterns of patients or their doctors, identified by the ImpACT questionnaire. It should be noted that for this study the Impact questionnaire was not used to evaluate psychological flexibility but was used considering every single item. Specifically, our results have shown that the behaviour of cancelling medical appointments was more apparent amongst subjects who contacted their medical specialist more often; in addition, this behaviour of cancelling medical appointments may have been further influenced by the community to which a patient belongs and reinforced particular by remaining in contact with other patients. Furthermore, patients who were afraid of medical staff becoming unwell and being unable to provide adequate treatment, were also more likely to cancel a medical appointment.

Some pattern of behaviours may reflect a dysfunctional use of experiential avoidance, as for i.e., the behaviour of asking reassurance may reflect an attempt to avoid or control the fear (item 10 “I have contacted my specialist doctor more often than usual for advice or reassurance.”), or the behaviour of search more often food, drink to avoid negative emotions (item 22 “I am eating more, drinking more alcohol and/or smoking more to cope with my emotions (e.g., boredom, apathy, sadness)). Patients who cancelled medical appointments were more likely to engage in such behaviours both during and after lockdown. Emotional problems and dysfunctional behaviour often manifest together and interact reciprocally over time, for example, depression often predicts the onset of eating disorders, and eating disorders can induce depression [36,37,38]. Similarly, poor health, functional impairment, stigma about obesity, and body image dissatisfaction, may all contribute to depression, while depression’s biological and psychosocial effects may lead to weight gain [39,40].

Overall, our results suggest that the decision to cancel a medical appointment is multifactorial, including psychological, sociodemographic status, relationship to health professional and isolation, and can be linked to several variables that affect patient behaviour in various ways.

Although we cannot clearly define what consequences cancelling medical appointments may have on disease progression, our results do suggest that such behaviour are related to an unhealthy and dysfunctional use of experiential avoidance and is a result of a fear of COVID-19.

According to ACT, psychological flexibility may be considered the key construct of psychological well-being and resiliency. Psychological flexibility enables an individual to alter their behaviour when it compromises their personal values, or is required to adapt to changing circumstances [41]. It involves facing difficult or uncomfortable situations by accepting, as opposed to avoiding, aversive thoughts, emotions and feelings. This flexibility means that personal values can remain uncompromised in the event of unpleasant situations [20]. In contrast, psychological inflexibility manifests as rigid psychological reactions that conflict with personal and societal values and can lead to undesirable behaviour. This often occurs when people attempt to avoid unwanted thoughts and feelings, and actually increases distress due to reducing one’s ability to connect with the present moment, and decreasing the likelihood of adhering to one’s personal values [42]. In such a context, people can feel overwhelmed by uncontrollable fear. Our data seems to be in line with previous data on Italian general population, in which the global psychological flexibility and four of its six sub-processes (self as context, defusion, values, committed action), mitigated the detrimental impacts of COVID-19 risk factors on mental health [20]; the psychological flexibility also mediating the decrease of the adverse effect of trait anxiety on COVID-19 distress, anxiety and depression. In contrast, embracing (rather than avoiding) inner discomfort and observing associated unhelpful thoughts, while also engaging in values-based action, increases resilience during adversity [21].

Our findings raise several questions about the delivery of care to patients who are afraid to attend a medical appointment. Fear may explain the higher rate of failing to attend follow-up appointments, particularly in cases where the patient has serious concerns about their illness and their continuity of care, or if the patient experiences high levels of distress or psychological issues, such as, anxiety, depression, or PTSD.

Furthermore, the results obtained from this study could be useful for implementing future guidelines or recommendations addressed to patients [43] and addressed to healthcare professionals [44].

## 5. Conclusions

The results of our research highlight that patients who tend to cancel medical appointments are also those who suffer from serious psychological problems, such as, PTSD or suicidal thoughts. Such individuals adopt avoidance behaviours, and unhealthy dysfunctional behaviours.

Post-traumatic stress symptoms have led to behaviours that attempt to control the aversive and fear-inducing situation (e.g., researching COVID-19 online), and other behaviours aimed at avoiding the aversive situation (e.g., excessive, and potentially harmful avoidance of social situations).

In situations of emergency, such as the SARS-CoV-2 outbreak, it may be useful to invest more in the psychological management of patients with pre-existing diseases, who may develop a heightened state of alertness and anxiety that could develop into PTSD. It is extremely important to identify the most vulnerable patients early on in order to provide adequate and effective support, this can be achieved by phone contact with especially anxious patients that refuse to attend appointments. It is also essential to support patients already diagnosed with PTSD, as the likelihood of clinical worsening in such scenarios as that of the SARS-CoV-2 outbreak is high. This would have direct repercussions on the patient’s quality of life, adherence to therapy, and on the progression of their disease.

During lockdown, many patients did not have sufficient access to psychotherapy and were left to manage their feelings without clinical support. As a result of this, pervasive dysfunctional behavioural patterns, such as, avoidance behaviours and reduced social contact were adopted, which increased the likelihood of cancelling medical appointments during the lockdown and after.

Understanding the underlying causes of cancelled medical appointments will allow time and valuable resources to be better utilised so that all patients may receive a higher standard of care.

Our findings offer an important contribution to our understanding of the psychological reaction of patients with pre-existing medical conditions during lockdowns and subsequent phases of a pandemic by describing the type and extent of distress experienced by patients and the impact of this on compliance with disease management requirements. Further study should aim to develop targeted and timely intervention to ensure the continued monitoring of especially vulnerable patients afflicted with medical disease.

## 6. Strengths and Limitations

The main strength of this study is our focus on patients with pre-existing medical conditions. As such, the majority of subjects were patients known to hospital services; this is a particularly relevant to online questionnaires, in which the true identity of respondents can be difficult to ascertain. However, the “snowball sampling strategy” as a recruitment method may be a limitation of the study due to possible selection bias for more compliant patients that are probably less likely to cancel medical appointments. A further limitation of the study may be linked to the external context (e.g., media, information provided by medical centre), that may have influenced the cancellation of appointments, an aspect that could be further evaluated in future studies. However, alternative methodologies were limited given the circumstances of an unexpected and sudden pandemic. A further limitation is the lack of any baseline evaluation of subjects before the pandemic, but this limitation is partially overcome by the comparison of the mean scores found in previous studies [7,23].

## Figures and Tables

**Table 1 ijerph-18-00340-t001:** Sociodemographic characteristics of individuals with pre-existing medical conditions. *N* is the number of respondents. Valid cases are the number of non-missing value.

	T1			T2		
	*n* = 758			*n* = 698		
	Valid Cases	Descriptive Statistics		Valid Cases	Descriptive Statistics	
**Gender *n* (%)**	758			697		
Male		191	(25.2)		258	(37.0)
**Age (Years) Mean (sd)**	758	50	(12)	693	52	(14)
**Number of children *n* (%)**	757			693		
0		298	(39.4)		272	(39.2)
1		172	(22.7)		141	(20.3)
2+		287	(37.9)		280	(40.4)
**Children age *n* (%)**	746			634		
[0–9]		82	(10.8)		17	(2.5)
[10–16]		80	(10.6)		22	(3.2)
[16+]		286	(37.7)		323	(46.6)
No children		298	(40.9)		272	(47.8)
**Education *n* (%)**	758			698		
Degree or Post degree		281	(37.1)		219	(31.4)
High school		341	(45.0)		344	(49.3)
Secondary school		118	(15.6)		100	(14.3)
Primary school		18	(2.4)		35	(5.0)
**Incidence of reported COVID-19 cases ***	754			690		
50.01–200		144	(19.1)		145	(21.0)
200.01–500		429	(56.9)		271	(39.3)
>500		181	(24.0)		274	(39.7)
**Home region *n* (%)**	754			690		
Northern Italy		564	(74.8)		511	(74.1)
Central Italy		67	(8.9)		45	(6.5)
Southern Italy		123	(16.3)		134	(19.4)
**Region of medical centre** ***n* (%)**	749			654		
Northern Italy		584	(78.0)		513	(78.4)
Central Italy		54	(7.2)		29	(4.4)
Southern Italy		111	(14.8)		112	(17.1)
**Marital status *n* (%)**	758			698		
Single		171	(22.6)		173	(24.8)
Married or living together		494	(65.2)		441	(63.2)
Separated or divorced		73	(9.6)		40	(5.7)
Widowed		20	(2.6)		44	(6.3)
**Employment status *n* (%)**	758			693		
Employed but working regularly		159	(21.0)		143	(20.6)
Employed but remote working		226	(29.8)		196	(28.3)
Unemployed		151	(19.9)		101	(14.6)
Retired		131	(17.3)		169	(24.4)
Stay-at-home spouse		79	(10.4)		74	(10.7)
Student		12	(1.6)		10	(1.4)
**Diagnosis *n* (%)**	758			698		
Cancer		205	(27.0)		232	(33.2)
Multiple sclerosis		104	(13.7)		73	(10.4)
Cardiovascular disease		75	(9.9)		44	(6.3)
HIV		29	(3.8)		80	(11.5)
Endocrine disease		38	(5.0)		-	
Rheumatic disease		120	(15.8)		57	(8.2)
Chronic migraine		38	(5.0)		39	(5.6)
Gynaecological disorders		32	(4.2)		28	(4.00)
Other		117	(15.5)		145	(20.8)

Notes: * per 100,000 population.

**Table 2 ijerph-18-00340-t002:** Questionnaire scores of surveyed subjects with pre-existing medical conditions. Valid cases are the number of non-missing value. CFQ-7: Cognitive Fusion Questionnaire; IES-R: Impact of Event Scale Revised, DASS-21: Depression Anxiety Stress Scale; IQR: Interquartile range.

	T1			T2		
	*n* = 758			*n* = 698		
	Valid Cases	Descriptive Statistics		Valid Cases	Descriptive Statistics	
**CFQ-7 Median (IQR)**	506	19 [13,14,15,16,17,18,19,20,21,22,23,24,25,26,27]		448	19 [12,13,14,15,16,17,18,19,20,21,22,23,24,25,26,27,28]	
**IES-R *n* (%)**	506			430		
Normal		223	(44.1)		188	(43.7)
Mild		79	(15.6)		80	(18.6)
Moderate		35	(6.9)		20	(4.7)
Severe		169	(33.4)		142	(33.0)
**DASS-21 Total Median (IQR)**	506	24 [14,15,16,17,18,19,20,21,22,23,24,25,26,27,28,29,30,31,32,33,34,35,36,37,38,39,40,41,42,43,44]		435	24 [10,11,12,13,14,15,16,17,18,19,20,21,22,23,24,25,26,27,28,29,30,31,32,33,34,35,36,37,38,39,40,41,42]	
**DASS-21 Depression *n* (%)**	506			435		
Normal		281	(55.5)		250	(57.5)
Mild		60	(11.9)		55	(12.6)
Moderate		99	(19.6)		73	(16.8)
Severe		29	(5.7)		26	(6.0)
Extremely Severe		37	(7.3)		31	(7.1)
**DASS-21 Anxiety *n* (%)**	506			435		
Normal		303	(59.9)		290	(66.7)
Mild		34	(6.7)		27	(6.2)
Moderate		90	(17.8)		59	(13.6)
Severe		24	(4.7)		19	(4.3)
Extremely Severe		55	(10.9)		40	(9.2)
**DASS-21 Stress *n* (%)**	506			435		
Normal		164	(32.4)		161	(37.0)
Mild		98	(19.4)		81	(18.6)
Moderate		146	(28.9)		99	(22.8)
Severe		51	(10.1)		46	(10.6)
Extremely Severe		47	(9.3)		48	(11.0)

**Table 3 ijerph-18-00340-t003:** Multivariate logistic regressions performed on T1 and T2 study populations. The dependent variables identify patients who cancelled a medical appointment during lockdown (T1) and after (T2).

	T1					T2				
	Univariable	Multivariable			Univariable		Multivariable			
		*n =* 475					*n =* 370			
	*p-*Value	OR	CI	*p-*Value	*p-*Value		OR	CI		*p-*Value
*Gender (ref. Male)*										
Female	0.006	1.067	0.550; 2.099	0.848		0.787				
*Incidence rate of reported COVID-19 cases per 100 000 population (ref. 50.01–200)*										
200.01–500	0.245	0.738	0.240; 2.284	0.595	0.869					
*>500*	<0.001	0.518	0.163; 1.622	0.257	0.635					
*Region of residence (ref. Northern Italy)*										
Central Italy	0.007	2.475	0.804; 7.613	0.114	0.108					
Southern Italy	<0.001	1.71	0.409; 7.138	0.463	0.926					
*Marital status (ref. Single)*										
Married or in a relationship	0.001	2.631	1.244; 5.702	**0.012**	0.804					
Separated or divorced	0.014	1.824	0.680; 4.878	0.230	0.162					
Widowed	0.127	5.189	0.997; 25.895	**0.045**	0.234					
*Number of children (ref. 0)*										
*1*	0.983	0.782	0.376; 1.617	0.508	0.710					
*2+*	0.036	1.095	0.552; 2.185	0.795	0.237					
*Employment status (ref. Employed and working regularly)*										
Employed but remote working	0.031	1.430	0.700; 2.956	0.329	0.656					
Unemployed	0.229	0.988	0.443; 2.202	0.976	0.420					
Stay-at-home spouse	0.001	0.879	0.301; 2.524	0.811	0.295					
Retired	0.647	0.868	0.363; 2.062	0.748	0.784					
Student	0.392	1.660	0.170; 10.904	0.623	0.156					
*Diagnosis (ref. HIV)*										
Cancer	0.029	0.669	0.171; 2.881	0.573	0.684		1.168	0.463; 3.068		0.747
Multiple sclerosis	0.081	2.645	0.621; 12.643	0.202	0.557		1.169	0.381; 3.603		0.784
Cardiovascular disease	0.242	1.019	0.228; 4.914	0.981	0.007		2.550	0.715; 9.286		0.150
Endocrine disease	0.593	1.605	0.327; 8.353	0.563	-		-			
Rheumatic disease	0.145	0.705	0.179; 3.055	0.626	0.484		0.504	0.144; 1.679		0.270
Chronic migraine	0.593	0.665	0.115; 3.910	0.647	0.085		1.510	0.383; 5.846		0.550
Gynaecological disorders	0.352	1.027	0.194; 5.649	0.975	0.550		0.728	0.146; 3.189		0.682
Other	0.220	1.504	0.379; 6.661	0.573	0.711		0.870	0.318; 2.425		0.786
*Region of medical centre (ref. Northern Italy)*										
Centre Italy	0.013	2.475	0.814; 7.778	0.114	0.313		2.709	0.574; 12.358		0.197
Southern Italy	<0.001	1.709	0.405; 7.159	0.463	<0.001		1.540	0.535; 4.393		0.418
*ImpACT (item 3): I feel confused and disoriented by the multitude of information I have heard about the Coronavirus (ref. Never true)*										
True	0.055	1.099	0.429; 2.995	0.848	0.806					
*ImpACT (item 4): I try to avoid thinking about the coronavirus (ref. Never true)*										
True	0.007	1.105	0.543; 2.321	0.786	0.004		0.999	0.409; 2.604		0.998
*ImpACT (item 7): I cannot tolerate being unable to leave the house anymore (ref. Never true)*										
True	0.001	0.865	0.402; 1.909	0.714	0.077		1.544	0.663; 3.800		0.326
*ImpACT (item 8): I feel more vulnerable because of my illness (ref. Never true)*										
True	0.001	1.030	0.460; 2.394	0.944	0.003		1.180	0.527; 2.768		0.694
*ImpACT (item 9): I fear not being able to continue medical treatment (ref. Never true)*										
True	<0.001	1.397	0.795; 2.451	0.244	<0.001		0.932	0.492; 1.749		0.827
*ImpACT (item 10): I have contacted my health care specialist more often than usual for advice or reassurance (ref. Never true)*										
True	<0.001	2.241	1.277; 3.947	**0.005**	<0.001		2.135	1.159; 3.972		**0.015**
*ImpACT (item 12): I fear doctors and nurses may get sick and not provide the attention I need (ref. Never true)*										
True	<0.001	2.169	1.252; 3.821	**0.006**	<0.001		3.409	1.835; 6.506		**<0.001**
*ImpACT (item 15): I am very worried about the future and this distracts me from the things I do in everyday life (ref. Never true)*										
True	<0.001	0.687	0.344; 1.372	0.285	0.009		0.659	0.320; 1.360		0.257
*ImpACT (item 19): I have kept in contact by phone with other patients (ref. Never true)*										
True	<0.001	2.126	1.250; 3.645	**0.006**	<0.001		0.857	0.484; 1.508		0.593
*ImpACT (item 20): I cannot believe this is happening to me (ref. Never true)*										
True	<0.001	1.367	0.803; 2.331	0.249		<0.001	1.268	0.686; 2.360	0.449	
*ImpACT (item 21): I feel more alone than usual (ref. Never true)*										
True	<0.001	1.963	1.062; 3.677	**0.033**		<0.001	0.897	0.450; 1.785		0.755
*ImpACT (item 22): I am eating more, drinking more alcohol or smoking more to cope with my emotions (e.g., boredom, apathy, sadness) (ref. Never true)*										
True	<0.001	1.728	1.034; 2.903	**0.037**		<0.001	2.568	1.427; 4.680		**0.002**
*ImpACT (item 23): I think stress worsens my symptoms (ref. Never true)*										
True	<0.001	1.465	0.685; 3.260	0.335		<0.001	1.742	0.744; 4.345		0.214
*ImpACT (item 24): I am worried about economic problems araising from the coronavirus outbrake (ref. Never true)*										
True	<0.001	1.603	0.743; 3.636	0.241		0.001	1.329	0.634; 2.924		0.462
*ImpACT (item 25): I have had suicidal thoughts (ref. Never true)*										
True	<0.001	0.863	0.390; 1.887	0.712		<0.001	0.593	0.222; 1.492		0.279
*ImpACT (item 26): Sometimes I remember past episodes so vividely, it is as if I am reliving it (ref. Never true)*										
True	0.018	0.578	0.322; 1.033	0.065		<0.001	1.345	0.728; 2.523		0.348
*ImpACT (item 27): I am using medicine to help handle anxiety (ref. Never true)*										
True	<0.001	1.180	0.613; 2.258	0.618		<0.001	1.556	0.744; 3.232		0.236
*ImpACT (item 28): I am doing more relaxation exercises or meditative practices to manage stress (ref. Never true)*										
True	0.143	0.667	0.394; 1.120	0.127		0.004	1.390	0.780; 2.487		0.264
*ImpACT (item 29): I think my religious faith is helping me (ref. Never true)*										
True	0.009	0.866	0.519; 1.447	0.582		0.013	1.098	0.601; 2.020		0.762
*ImpACT (item 30): I am satisfied with my life (ref. Never true)*										
True	0.769					0.190	0.793	0.174; 4.004		0.770
*ImpACT (item 31): Living in this situation has helped me to see life through different eyes (ref. Never true)*										
True	0.250					0.172	0.840	0.269; 2.824		0.770
*Trusted sources of information (ref. Other)*										
Relatives or friends	0.017	1.499	0.705; 3.162	0.288		0.824				
*Trusted sources of information (ref. Other)*										
Doctor, nurse, psychologist or other health care professionals	<0.001	0.847	0.471; 1.511	0.577		0.344				
*Trusted sources of information (ref. Other)*										
Instant messaging apps e.g., WhatsApp, Telegram	<0.001	0.383	0.106; 1.295	0.131		<0.001	3.275	0.598; 18.546		0.170
*Trusted sources of information (ref. Other)*										
Web sources	0.004	1.448	0.771; 2.715	0.248		0.002	1.640	0.809; 3.342		0.169
*Trusted sources of information (ref. Other)*										
Social media	0.002	0.383	0.106; 1.295	0.131		<0.001	0.564	0.202; 1.478		0.257
*Trusted sources of information (ref. Other)*										
Ministry of health, ISS	0.004	0.672	0.347; 1.303	0.237		0.577				
*Psychologically supported in the past (ref. No)*										
Yes	<0.001	1.071	0.605; 1.893	0.813		0.010	0.803	0.421; 1.514		0.500
*Psychological support is useful (ref. No)*										
Yes	<0.001	0.892	0.484; 1.632	0.713		<0.001	1.282	0.634; 2.595		0.488
*My life has changed from that of before (ref. No)*										
Yes	0.015	1.808	0.636; 5.629	0.282		0.403				
*CFQ-7*	0.003	0.958	0.916; 1.000	0.052		0.006	0.978	0.933; 1.024		0.350
*IES-R*	<0.001	1.025	1.002; 1.048	**0.032**		<0.001	1.008	0.985; 1.032		0.478
*DASS-21*	0.001	0.999	0.982; 1.016	0.890		<0.001	1.000	0.982; 1.018		0.986
*R^2^ Tjur*		0.281					0.217			
Hosmer–Lemeshow test		χ^2^ (8) = 6.859 *p* = 0.552					χ^2^ (8) = 10.542 *p* = 0.229			

*n* refers to the complete cases. Figures in boldface reflect statistically significant coefficients. CI: Confidence Interval, OR: Odds Ratio, CFQ-7: Cognitive Fusion Questionnaire; IES-R: Impact of Event Scale Revised, DASS-21: Depression Anxiety Stress Scale, ISS: Italian national health institute.

## Data Availability

Data available on request due to ethical restrictions.

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
