# Peer review of "The Impact of the SARS-CoV-2 Outbreak on the Psychological Flexibility and Behaviour of Cancelling Medical Appointments of Italian Patients with Pre-Existing Medical Condition: The “ImpACT-COVID-19 for Patients” Multi-Centre Observational Study"

_ijerph, 2021, doi:10.3390/ijerph18010340_

Round 1

Reviewer 1 Report

1- English language and style for introduction need to be improved.

2- In line 158, how can the size of respondents' homes affect the results?

3- There is not any baseline evaluation, and this is a significant flaw of the study that needs to be justified or more studies should be included.

4- how compliant patients are distinguished from non-compliant patients?

Author Response

1- English language and style for introduction need to be improved.

The article was submitted and reviewed by an expert native English speaker

2- In line 158, how can the size of respondents' homes affect the results?

Thank you for raising this point. We removed the sentence “the size of the home (number of rooms, garden, terrace)", as it is not a variable used in multivariate regression, and therefore may confuse the reader.

3- There is not any baseline evaluation, and this is a significant flaw of the study that needs to be justified or more studies should be included.

Given the exceptionality of the COVID event, it was not possible to carry out a baseline assessment.

4- how compliant patients are distinguished from non-compliant patients?

We apologize but we are not sure to have well understood the issue of the compliant and non-compliant patients: If the question relates to compliance with the questionnaires, we can say that the patients who completed the questionnaire are identified as "Valid case" (in the table 1 and table 2); if it is meant "compliance with medical treatments", this was not an objective of the study.

Reviewer 2 Report

The study is interesting and original. I have two main points of concern, one conceptual and one methodological. Addressing these will, in my opinion,  increase the scientific soundness and merit of the paper.

Conceptual: The authors' concern is patients'  cancelling of medical appointments. However, as stated by the authors,  medical centres also reach out to patients to cancel or postpone appointments. Thus, patients' cancelling of appointments happen in a context where this is common practice, and where patients are likely to know that seeking medical centres may actually constitute an increased risk of contracting Covid-19. In view of this, it seems difficult to claim that the cancelling of medical appointments happen due to low levels of 'psychological flexibility'. In the Introduction (and possibly limitation) section(s), it would be important to see the authors' reasoning about this isssue.

Methodological: The ImpACT questionnaire was developed for this study, and no measurement properties are reported. Thus, there seems to be no stated reasons (other than theoretical) why one should trust scores on this instrument to be a good reflection of the theoretical concepts. In the Methods section, it would be important to see the authors' empirical reasons for why they trust this instrument to provide a sound assessment of 'psychological (in-)flexibility'.

Minor issues:

line 136: recruiting from the general public sounds odd, given that this is a study of patients with pre-existing conditions?

Some sentences are difficult to penetrate, e.g., line 188: What does '...guided more by relatively inflexible verbal networks than by contacted environmental contingencies' really mean?

lines 173-216: Lots of theory concerned with the ImpACT measure is provided (should be reduced), whereas information about its measurement properties is missing (should be increased). Why report scale composition when the analysis uses single items as predictors?

Results are well presented, while the Discussion may incorporate some of the issues addressed in this review.

Author Response

The study is interesting and original. I have two main points of concern, one conceptual and one methodological. Addressing these will, in my opinion,  increase the scientific soundness and merit of the paper.

Conceptual: The authors' concern is patients'  cancelling of medical appointments. However, as stated by the authors,  medical centres also reach out to patients to cancel or postpone appointments. Thus, patients' cancelling of appointments happen in a context where this is common practice, and where patients are likely to know that seeking medical centres may actually constitute an increased risk of contracting Covid-19. In view of this, it seems difficult to claim that the cancelling of medical appointments happen due to low levels of 'psychological flexibility'. In the Introduction (and possibly limitation) section(s), it would be important to see the authors' reasoning about this isssue.

Thank you for giving us the opportunity to better reformulated the paragraph as follows:  (line 146-149)" It should be noted that many patients have not cancelled their scheduled medical appointments, despite the influence of outside information, including social media (sometimes very conflicting or confusing) and institutions (ministry of health, medical centers)..”

(line 157-160) “As yet, there have been no published studies on the behaviour of cancelling medical appointments of patients with pre-existing medical condition during the SARS-CoV-2 outbreak, and especially none that consider the psychosocial factors influencing the decision to cancel appointments regularly scheduled by medical centers.”

We have also added a part to better describe the reason it may also be useful to evaluate psychological flexibility to better understand the behavior of cancelling medical appointments. Therfore, we haveadded the following part in the text: (line 161-178) “The behavior of cancelling medical appointments can be linked to different psychological functions that occur in relation to the external context (decisions of the ministry, medical centers, social and health contexts) and in relation to the internal context (thoughts, emotions, physical sensations). Therefore, in some cases this behaviour can be functionally adaptive to the context or in other cases it can be highly dysfunctional to the context and have negative consequence on one's own and others' health. The ability to adapt is closely linked to the degree of psychological flexibility, defined as acting in accordance with personal goals and values, in the presence of potentially interfering thoughts and feelings, and with a greater appreciation of what their current situation or context allows (Hayes, Strosahl, & Wilson, 2012).

Moreover, recent studies have shown that psychological flexibility mitigated the detrimental impacts of the pandemic on mental health, peritraumatic distress, anxiety, depression, insomnia and facets of psychological inflexibility exacerbated the impact of these risks (Pakenham et al., 2020; Landi et al, 2020 ; McCracken et al. 2020;).

We added some consideration about the limits of the study: “A further limitation of the study may be linked to the external context (e.g. media, information provided by medical center), that may have influenced the cancellation of appointments, an aspect that could be further evaluated in future studies.”

Methodological: The ImpACT questionnaire was developed for this study, and no measurement properties are reported. Thus, there seems to be no stated reasons (other than theoretical) why one should trust scores on this instrument to be a good reflection of the theoretical concepts. In the Methods section, it would be important to see the authors' empirical reasons for why they trust this instrument to provide a sound assessment of 'psychological (in-)flexibility'.

Thank you for raising this point. We have removed the last two paragraphs of the ImpACT questionnaire, as they could create confusion in the reader: in the multivariate analysis we have carried out in this article, the questionnaire was not used as a whole to evaluate psychological flexibility but was used considering every single item

Minor issues:

line 136: recruiting from the general public sounds odd, given that this is a study of patients with pre-existing conditions?

Thank you for raising this point, as the term "general public" can be misunderstood, and for this reason we have rephrased the sentence as follows: “A ‘snowball sampling strategy’, focusing on recruiting other participants with pre-existing medical conditions of mainland Italy then followed.

Some sentences are difficult to penetrate, e.g., line 188: What does '...guided more by relatively inflexible verbal networks than by contacted environmental contingencies' really mean?

Thank you for raising this point. We reformulated these  sentences “In contexts that foster such fusion, human behavior is guided more by relatively inflexible verbal networks than by contacted environmental contingencies”, as follows: when cognitive fusion increases, human behavior is less sensitive to environmental contingencies.”

lines 173-216: Lots of theory concerned with the ImpACT measure is provided (should be reduced), whereas information about its measurement properties is missing (should be increased). Why report scale composition when the analysis uses single items as predictors?

I confirm what we wrote at your first observation on the methodological part (We have removed the last two paragraphs of the ImpACT questionnaire, as they could create confusion in the reader: in the multivariate analysis we have carried out in this article, the questionnaire is not used as a whole to evaluate psychological flexibility but is used considering every single item)

Results are well presented, while the Discussion may incorporate some of the issues addressed in this review.

We introduce this sentence in the discussion: “It should be noted that for this study the Impact questionnaire is not used to evaluate psychological flexibility but is used considering every single item.”

Reviewer 3 Report

I commend the authors for their study on determinants of non-adherence to medical appointments.

I have some questions and comments.

  1. With the advent of the pandemic and the number of cases spiraled out of control, many health facilities have had to cancel routine appointments, including non-emergency/elective surgery depending on the local protocol. Therefore,
  • What is the proportion of provider-initiated appointment cancellations, in T1 and T2?
  • What is the proportion of patient-initiated appointment cancellations, in T1 and T2?
  • It was reported that many cancellations happened, after over-the-phone consultation with a specialist. In this era of telemedicine, should we consider over-the-phone consultations equivalent to appointment cancellations?
  1. Patient- initiated medical appointment non-attendance is two types: some are cancellations, some others could be no-shows. What proportion were actual cancelations, and what proportion were No-Shows/Missed appointments?
  2. What was the timeline pattern of cancellations, within 24 hours, 48 hours, 5 days, one week (early/late cancellations), etc.?
  3. What is the validity and reliability of each of the survey instruments, both established and newly developed ones?
  4. What is the variance of each of the significant independent variables/factors that explain the outcome variable (patient-initiated cancellation of medical appointments)?
  5. I suggest that only the statistically significant variables be summarized in Table 3, all the statistically insignificant variables can be mentioned in the narrative.

Author Response

I commend the authors for their study on determinants of non-adherence to medical appointments.

I have some questions and comments.

  1. With the advent of the pandemic and the number of cases spiraled out of control, many health facilities have had to cancel routine appointments, including non-emergency/elective surgery depending on the local protocol. Therefore,
  • What is the proportion of provider-initiated appointment cancellations, in T1 and T2?
  • What is the proportion of patient-initiated appointment cancellations, in T1 and T2?

We apologize but we do not have the data to calculate the proportions that you indicated, because the aim of the study was to evaluate the psychological status and related psychosocial factors of patients who reported having canceled medical appointments.

We also believe that these data are uninformative for the study objectives, as we refer to patient-initiated medical appointments.

2. It was reported that many cancellations happened, after over-the-phone consultation with a specialist. In this era of telemedicine, should we consider over-the-phone consultations equivalent to appointment cancellations?Patient- initiated medical appointment non-attendance is two types: some are cancellations, some others could be no-shows. What proportion were actual cancelations, and what proportion were No-Shows/Missed appointments?

We apologize but we do not have this  data. We included only the data reported by the patient on having autonomously canceled the appointment of the medical examination. Therefore we don’t have data about No-Shows/Missed appointments.

3. What was the timeline pattern of cancellations, within 24 hours, 48 hours, 5 days, one week (early/late cancellations), etc.?

We apologize but we do not have this  data. We included only the data reported by the patient on having indipendently canceled the appointment of the medical examination.

4. What is the validity and reliability of each of the survey instruments, both established and newly developed ones?

Thank you for suggestion. We added in the text values of Cronbach alpha for each scale used. “In our study, IES-R showed good internal consistency in both rounds of questionnaire (0.94 in T1 and 0.95 in T2).” “Cronbach’s alpha for the total scales was 0.96 in both T1 and T2.”; “The test showed good internal consistency, with Chronbach’s alpha of 0.91 in T1 and T2.”

5. What is the variance of each of the significant independent variables/factors that explain the outcome variable (patient-initiated cancellation of medical appointments)?

The variance of the estimates was expressed through the use of confidence interval.

6. I suggest that only the statistically significant variables be summarized in Table 3, all the statistically insignificant variables can be mentioned in the narrative.

Thank you for suggestion. In this case we prefer to keep all the variables listed in the table 3.

Round 2

Reviewer 1 Report

The responses are acceptable. Thanks